# Botulinum Toxin Injection for the Treatment of Upper Esophageal Sphincter Dysfunction

**DOI:** 10.3390/toxins14050321

**Published:** 2022-04-30

**Authors:** Pengxu Wei

**Affiliations:** 1Beijing Key Laboratory of Rehabilitation Technical Aids for Old-Age Disability, Key Laboratory of Neuro-Functional Information and Rehabilitation Engineering of the Ministry of Civil Affairs, National Research Center for Rehabilitation Technical Aids, No. 1 Ronghuazhong Road, Beijing Economic and Technological Development Zone, Beijing 100176, China; pengxuwei@buaa.edu.cn; 2Key Laboratory of Biomechanics and Mechanobiology (Beihang University), Ministry of Education, School of Biological Science and Medical Engineering and Beijing Advanced Innovation Center for Biomedical Engineering, Beihang University, 37# Xueyuan Road, Haidian District, Beijing 100083, China

**Keywords:** dysphagia, upper esophageal sphincter, cricopharyngeal muscle, botulinum toxin, injection, electromyography, ultrasound

## Abstract

Dysphagia associated with upper esophageal sphincter (UES) dysfunction remarkably affects the quality of life of patients. UES injection of botulinum toxin is an effective treatment for dysphagia. In comparison with skeletal muscles of the limb and trunk, the UES is a special therapeutic target of botulinum toxin injection, owing to its several anatomical, physiological, and pathophysiological features. This review focuses on (1) the anatomy and function of the UES and the pathophysiology of UES dysfunction in dysphagia and why the entire UES rather than the cricopharyngeal muscle before/during botulinum toxin injection should be examined and targeted; (2) the therapeutic mechanisms of botulinum toxin for UES dysfunction, including the choice of injection sites, dose, and volume; (3) the strengths and weaknesses of guiding techniques, including electromyography, ultrasound, computed tomography, and balloon catheter dilation for botulinum toxin injection of the UES.

## 1. Introduction

Dysphagia (swallowing disorders) is a common problem for many diseases, including stroke (8.1–80%), Parkinson’s disease (11–81%), traumatic brain injury (27–30%), community-acquired pneumonia (91.7%) [1], Alzheimer’s disease (85.9%) [2], and mental illness (9–42%) [3]. Dysphagia causes serious complications, including malnutrition, weight loss, aspiration pneumonia, premature mortality, and airway obstruction (e.g., choking, asphyxia) and affects people of all ages. The prevalence of dysphagia ranges from 5% to 72% in the community-dwelling elderly [4], and 20–80% of infants in high-risk neonatal intensive care units present feeding concerns during infancy [5].

The dysfunction of the upper esophageal sphincter (UES) impedes swallowing function recovery in patients with dysphagia [6]. A botulinum toxin injection is a procedure for the treatment of UES dysfunction [7]. This review introduces the pathophysiology of UES dysfunction in dysphagia, and the therapeutic mechanisms of botulinum toxin injection for UES dysfunction, including the choice of injection sites, dose, and volume, and the strengths and weaknesses of guiding techniques for Botulinum toxin injection in the UES, such as electromyography (EMG), ultrasound, computed tomography (CT), and balloon catheter dilation.

## 2. UES Anatomy and Function

As a segment located between the pharynx and esophagus, the UES is a high-pressure zone that prevents reflux into the airway and the entry of air into the digestive tract [8]. In healthy individuals (20–34 years old), the pressure in the upper esophagus is 20.39 ± 15.06 mm Hg above the atmospheric pressure [9].

The muscular components of the UES are confusingly introduced. The UES was proposed to be composed of three muscles, namely, the inferior pharyngeal constrictor, the cricopharyngeus, and the upper portion of the cervical esophagus in a cranial-to-caudal direction [10,11]. Under physiological conditions, the cricopharyngeus, inferior pharyngeal constrictor, and possibly the infracricoid esophagus were considered to generate a basal tone, and the tension was relieved during swallowing by active relaxation, which occurred mainly from the cricopharyngeus [12,13,14]. The cricopharyngeus was considered to be the main part to maintain the tone of the UES because this muscle presents EMG signal fluctuations associated with UES pressure changes, has a continual basal tone, and relaxes during swallowing [10,13,14]. The peak UES pressure was detected at the level of the inferior pharyngeal constrictor [15,16,17] or the cricopharyngeus [18].

The cricopharyngeus was synonymous with UES because it was the main part of the UES [9,19], and cricopharyngeal achalasia/dysfunction referred to the inadequate/failure opening of the cricopharyngeal muscle [20,21,22,23,24,25,26].

However, recent studies indicate that the inferior pharyngeal constrictor consists of two parts, namely, the cricopharyngeus and thyropharyngeus [27,28] (Figure 1). The names of the two muscles indicate their anatomical origins, namely, the cricoid cartilage and thyroid cartilage. Thus, an accurate description is to replace “inferior pharyngeal constrictor” with “thyropharyngeus” in the aforementioned introduction of the UES, or at least, to clearly define the anatomical scope of the “inferior pharyngeal constrictor”.

The UES pressure is generated by myogenic activity and passive elasticity of the tissues. The cricopharyngeus is a striated muscle consisting of thin muscle fibers (25–35 μm, predominantly slow-twitch type I) and 40% endomysial elastic connective tissue, which are connected together to form a muscular net [10].

The thickness/width of the cricopharyngeal muscle and cervical esophagus are important factors for botulinum toxin injection. The wall of the cervical esophageal junction consists of an outer longitudinal layer and an inner circular layer of muscle fibers. The two layers ascend and continue with the tendinous band near the mid-posterior cricoarytenoid level. In cadavers, the thicknesses of the outer layer in the anteromedian, anterolateral, and lateral parts are 0.35 ± 0.16, 1.85 ± 0.45, and 0.73 ± 0.39 mm, and those of the circular layer in the anteromedian, anterolateral, and lateral parts are 0.70 ± 0.35, 0.62 ± 0.22, and 0.74 ± 0.22 mm, respectively [28]. In young healthy subjects measured using endoluminal sonography, the right-side width of the muscle layer of the cricopharyngeal muscle (the hypoechoic musculature between hyperechoic adventitia and mixed-echoic mucosa) is 2.7 ± 0.6 mm, and the left-side width is 2.8 ± 0.6 mm [9]. The full width of the opened cricopharyngeal muscle (including the musculature, adventitia, and mucosa) during a 10 mL-water swallow was 4.57 ± 1.6 mm, as determined using transcutaneous ultrasound scanning in 20–56-year-old healthy subjects [29]. Taken together, the muscle layer of the cricopharyngeal part of the UES (2.7–2.8 mm) in vivo is more than half of its full width (4.57 ± 1.6 mm), and the inner circular muscle layer is thinner or is similar to the outer longitudinal layer in the anterolateral (0.62 vs. 1.85 mm, in cadavers) or lateral parts (0.74 vs. 0.73 mm, in cadavers). The circular muscle layer, rather than the longitudinal layer, contributes to the UES opening deficits and is the target of intervention (Figure 2).

The cervical esophagus may also be an injection target for some patients (see next section). In patients with gastroesophageal reflux, the anterior wall thickness of the cervical esophagus (defined as the distance between the adventitia and mucosa, which is thicker than the width of the muscle layer alone) was 2.3 ± 0.4 mm (1.6–4.0 mm), as measured by transcutaneous ultrasound [30], similar to the normal cervical esophagus (2.9 ± 0.2 mm or 2.8 ± 0.4 mm with different methods) [31] but thinner than the full width/thickness of the cricopharyngeal muscle (4.57 ± 1.6 mm).

In healthy subjects, five phases of UES activities during swallowing were proposed based on synchronized manometry and barium study (videofluoroscopic swallow study, VFSS). In phase I, tonic UES contraction is inhibited. The falling of UES pressure to zero is defined as UES relaxation, which appears 0.1 s before the VFSS-demonstrated UES opening and is independent of the bolus volume. Phase I is followed by the elevation of the hyoid bone and larynx (phase II), which provides an active opening force of the UES. In phase III, the moving bolus expands the UES because of UES elasticity and bolus pressure. After the bolus passes, the elasticity of the UES leads to UES closure (phase IV), followed by active UES muscle contraction (phase V) [32]. Moreover, manometry-detected UES relaxation occurs before fluoroscopic UES opening, but manometry-recorded UES contraction occurs 0.05–0.1 s earlier than the fluoroscopic closing of the UES [16]. Therefore, the abovementioned phase V is earlier than phase IV.

The deactivated cricopharyngeal EMG signal matches the decreased UES pressure during UES opening. Notably, in comparison with a small-volume (2 or 5 mL) bolus, swallowing a large (10 or 20 mL) bolus leads to delayed peak suprahyoid muscle activity, shifting from before to following peak UES opening [33]. That is, the passive opening caused by the elasticity of the UES and the pressure of the bolus occurs earlier than the force generation by suprahyoid muscle contractions when swallowing a large bolus. Therefore, stretching (e.g., dilating the UES with an inflated balloon) may be considered to correct decreased elasticity as a supplementary intervention to botulinum toxin injection.

## 3. Length of UES Dysfunction

During a barium study, the UES opening can be viewed at a level of approximately 5 mm below the inferior surface of vocal folds in healthy subjects [34], whereas the “cricopharyngeal bar” or “jet effect” indicates UES dysfunction [35] and related inefficiency of bolus transit through the UES [6].

The UES generates a high-pressure zone with a length of 30–45 mm measured by manometry, and the main part is believed to be the cricopharyngeal muscle with a length of 20–40 mm [36], which is much longer than the 0.75-cm length of the horizontal fibers of the cricopharyngeal muscle [37]. The cervical esophagus portion of the UES has a length of 1–2 cm, presenting a transitional anatomic structure between the cricopharyngeal muscle and the adjacent lower section of the esophagus and demonstrating unique histologic features and the presence of circular muscle fibers [38]. This cervical esophagus segment is located between C6 and C7 [38], at a level just below the cricopharyngeal muscle and the cricopharyngeal bar (reported at various levels, from C4 to C6) [39,40].

The UES narrowing may be longer than the aforementioned length. The narrowed segment below the cricopharyngeus may be “several centimeters long” in 60% of the cases and consists mainly of circular muscle fibers [38]. The studies also find that the esophageal segments below the level of an ordinary UES length may present narrowing. For example, histological abnormalities in the cervical esophagus 2 cm below the UES (called sub-UES) or in an even longer region may be observed in patients with dysphagia and/or endoscopically-observed narrow esophagus [41]. The lower border of the UES narrowing may be at the level of the first thoracic vertebra in patients with dysphagia [42]. The barium studies find that patients with dysphagia may present narrowing in the upper esophagus, mid esophagus, or entire esophagus [43]. Thus, the lowest level of UES dysfunction (or more exactly, the narrowed segment) should be determined before botulinum toxin injection.

The UES pressure during rest may be mainly generated by several reflexes, responses, and muscle mechanics instead of specific tone-generating neural circuits in the brainstem [10]. Some patients with dysphagia present hypertrophy of the cricopharyngeus muscle [44], which is characterized by a thick appearance with atrophy, fibrous, and chronic inflammation of the muscle [45].

In addition to the relaxation of UES muscles, forward movement of the larynx by the contraction of suprahyoid muscles contributes to UES opening. Although the forward movement of the hyoid and larynx exerts a pulling-forward force to open the UES, a backward fixation force is also needed. Otherwise, the UES will be pulled forward as a whole rather than be opened. This seems to be overlooked in the literature. In comparison with the forward movement of the hyoid–larynx complex, the posterior pharyngeal wall at the level of the cricoid cartilage presents a much shorter displacement in the anterior-posterior direction (hyoid bone, 12.8 ± 3.7 mm; larynx, 9.7 ± 3.3 mm; posterior pharyngeal wall, 5.6 ± 2 mm) during swallowing [46]. This finding supports the existence of a backward fixation mechanism of the posterior UES wall during UES opening, although the study results on the presence of tight connections between the posterior pharyngeal wall and prevertebral fascia are not consistent [47]. A much longer displacement of the posterior pharyngeal wall at the level of the cricoid cartilage is observed in the vertical direction than in the anterior-posterior direction (23.4 ± 4.1 mm vs. 5.6 ± 2 mm) during swallowing [46], indicating little effects of the movement restricting mechanism on the posterior pharyngeal wall in the vertical direction.

The median pharyngeal raphe is the fixation structure of the posterior pharyngeal wall, thereby exerting a backward force during UES opening. As introduced in the textbook, the inferior, superior, and middle pharyngeal constrictors insert posteriorly into the median pharyngeal raphe, which is a prevertebral midline fibrous band [27]. However, anatomical variations exist in the structure of the median pharyngeal raphe. A study in human cadavers found that in most cases (47%), the pharyngeal raphe was only located between the inferior constrictor muscles; in 40% of the cases, the raphe was only located between the superior and middle constrictor muscles and was absent between the inferior pharyngeal constrictors; in 13% of the cases, the raphe extended through all the three constrictors. Additionally, the pharyngeal raphe was an interrupted rather than a continuous line in most cases [48]. A study found that the cricopharyngeal muscle, which originates from the lowermost lateral edges of the cricoid lamina on each side, demonstrated no median pharyngeal raphe [38]; in this condition, the possible maximum extent of the median pharyngeal raphe includes only the superior and middle pharyngeal constrictors and the thyropharyngeus muscle (upper part of the inferior constrictor), and the cricopharyngeus muscle has no such backward fixation.

The anatomical variations in the pharyngeal raphe affect UES opening. The opening of the lower part of the UES, which consist of the cricopharyngeus and the adjacent lower part, can be difficult for subjects without the median pharyngeal raphe at this level, thereby lacking a pull-backward force. This configuration may explain the UES opening deficits for such patients.

## 4. Factors Influencing Effects of Botulinum Toxin Injection

An intramuscular injection of a botulinum toxin preparation blocks acetylcholine release from the nerve endings at the neuromuscular junction (i.e., the motor endplate) of the muscle. Different botulinum toxin products, such as Botox (Allergan, Irvine, CA, USA), Dysport (Ipsen, Slough, UK), Xeomin (Merz Pharma, Frankfurt am Main, Germany), and Hengli (Lanzhou Biological Products Institute, Lanzhou, China), have various manufacturing processes and formulations. Such differences lead to various interactions between the product and the tissue injected. Thus, the dosing and performance of these products are uninterchangeable, and it should be cautious to use these products in simple dose ratios [49,50].

The local spread/diffusion characteristics of the botulinum toxin may affect the botulinum toxin injected into the UES. Several terms, including diffusion, spread, and migration, have been used to describe the physical movements of the toxin from the injected site to other areas of the body. Botulinum toxin moves to areas other than the injected site, resulting in the local, distal, and systemic effects of the therapy through mechanisms including molecular dispersion, neuro-axonal transport, or hematogenous transport [51]. Here, local spread/diffusion refers to a passive dispersion process by which botulinum toxin moves to adjacent areas from the injected site.

Usually, a more limited extent of local spread/diffusion is preferred because the injected toxin has a lower chance of paralyzing muscle fibers near the injection site. However, a greater extent of spread/diffusion of botulinum toxin may effectively relieve UES narrowing. For instance, if an injection presents a 10 mm–radius scope of spread/diffusion around the injected site, it can lead to greater inhibition of the hyperactive UES activities than an injection leading to 5 mm–radius spread/diffusion. More localized effects (less extent of spread/diffusion) of a product are favorable for large muscles to avoid influencing untargeted muscles near the injection sites. However, a higher extent of spread/diffusion is preferred for UES injection of botulinum toxin.

The role of spread/diffusion in UES botulinum toxin injection is more important than the injection into large-volume muscles. For UES dysfunction, the injection targets the circular muscle layers of the UES wall. Such circular muscle layers are much smaller than limb muscles, and a higher degree of spread/diffusion can result in a more extensive distribution of botulinum toxin inside the UES wall. In the cricopharyngeus and cervical esophagus portions of the UES, the distribution of motor endplates does not form a motor endplate band but presents a scattered pattern [52]. Instead of toward motor endplate enriched zones, the injection sites are usually selected on the basis of the local anatomy of the UES. Under normal conditions, the uppermost part of the esophagus inclines slightly to the left side of the neck [53]. When transcutaneous ultrasonography is used for scanning, the majority of the cervical esophagus can be seen from the left side rather than from the right side (left lateral 2/3 vs. right lateral 1/3) [31]. Thus, for percutaneous botulinum toxin injection, an approach through the left side of the neck is easier to reach the target muscle layer in the anterolateral wall of the UES. Under such conditions, the injected toxin needs to spread to reach as many scattered motor endplates as possible and thus effectively inhibit UES hyperactivity. Therefore, spread/diffusion is important for UES botulinum toxin injection in comparison with large-volume muscles.

Different botulinum toxin products present various local spread/diffusion characteristics. When using the same volume, Botox presents a smaller diffusion area than Dysport [54] and Hengli [55]. Similarly, when injected into the mouse leg, Botox, Dysport, and Xeomin present slightly different spread/diffusion rates in muscles near the injected site, though the difference is not statistically significant [56]. Such a mild difference may not lead to obviously dissimilar effects for a large-volume muscle, e.g., biceps brachii or soleus, but may influence the effect of botulinum toxin injection into a thin musculature such as the UES. A higher diffusion capacity of a botulinum toxin product may spread to larger areas in the UES wall and thus improve the therapeutic effects.

Another point is the length of the UES dysfunction. For a relatively long UES narrowing, more injection sites should be considered. When a single bolus is injected into one site in the musculature of the UES, its spread/diffusion is unlikely to cover the entire length of the affected segment. In this condition, the total dose can be divided into several parts [51], and botulinum toxin can be injected into more than one site along the long axis of the UES.

The fluid volume and force of injection are proposed to be key factors for toxin distribution within the muscle [57]. When using an MRI to compare the distribution of injected botulinum toxin or normal saline into healthy biceps brachii muscles and spastic biceps brachii, the spread/diffusion in the healthy muscle presents a long and thin layer parallel to the muscle fibers, whereas the spread/diffusion in the spastic muscle exhibits a short and thicker layer, indicating that the spasticity changes the spread/diffusion process after injection [58]. Therefore, considering the muscular hypertonicity of the UES narrowing, more injection sites may be considered to make botulinum toxin spread over broader areas. Additionally, the MRI study [58] also found that an increased injection volume of normal saline results in a thicker layer but with a consistent length. Similarly, after injecting 10 U of botulinum toxin diluted with 0.1 or 0.5 mL of normal saline into the gastrocnemius muscles of rabbits, the higher dilution volume resulted in a better inhibition effect compared with a lower dilution volume [59]. A large volume (2 U/0.1 mL) also resulted in greater diffusion and a larger affected area than a smaller volume (2 U/0.02 mL) after injecting 5 U of botulinum toxin A for the treatment of dynamic forehead lines [60]. Taken together, an increase in the volume of injection for a given site may enhance the spread/diffusion process and effects of the toxin, possibly because of the higher pressure generated by a larger volume.

## 5. Guiding Techniques for Botulinum Toxin Injection

Under most conditions, the targets of UES botulinum toxin injection are the cricopharyngeus (level with the cricoid cartilage) and the upper part of the cervical esophagus (posterior to the trachea, approximately between C6 and C7 [38]). A cricopharyngeal bar viewed by a barium study is usually located between C4 and C6 [39,40]. The cervical esophagus in the lower neck deviates to the left and is closer to the carotid sheath and thyroid gland on the left side than on the right side [53].

To accurately insert the needle into a target site in the thin UES wall and bypass its adjacent organs, appropriate guiding techniques are needed. CT [42,61], endoscope [21,24,62,63,64,65,66,67,68,69,70,71,72,73,74,75,76,77], ultrasound [78,79,80,81,82,83], and EMG [7,62,63,66,73,78,80,81,84,85,86,87,88,89] have been reported as guiding methods.

When an injection is guided by EMG, the muscle activities of the thyropharyngeus (introduced as the inferior pharyngeal constrictor in some studies) and cricopharyngeus can be viewed and should be differentiated. The appearance of EMG signals only indicates the entrance of the needle tip into muscle tissues and cannot ensure that it is the cricopharyngeus. Under normal conditions, the thyropharyngeus is electrically silent during rest but active during swallowing, whereas the cricopharyngeus presents tonic activity during rest but is relatively silent during swallowing [85,90]. For healthy adults, the silent duration (i.e., electromyographic pause) of the cricopharyngeus during swallowing ranges from 300 ms to 600 ms [88]. One week after botulinum toxin injection, the shortened duration (132 ± 96.7 ms) in patients with UES dysfunction was corrected to 375 ± 89.2 ms [89].

Schneider et al. [63] applied a rigid esophagoscope to identify the bulge of the cricopharyngeus and then inserted a hooked wire electrode to identify hyperactive EMG signals of the muscle without a relaxation period. Transcutaneous approaches to inserting the EMG electrode have been applied [7,62,66,73,78,80,81,84,85,86,87,88,89]. The larynx can be manually rotated, and the needle can be inserted near the inferior border of the cricoid cartilage and then be advanced posteromedially, following the contour of the cricoid cartilage [85,86]. The EMG needle electrode can also be inserted at the level of the cricoid cartilage, 1.5 cm posterior to its palpable lateral border, in the posteromedial direction [7,89,90]. When using the transcutaneous approach, the patient may be asked to vocalize to avoid misplacement of the needle into the intrinsic laryngeal muscles and to tense the neck and tilt the head to avoid misplacement in the strap or in the paraspinal muscles [73,86]. When inserting and advancing the EMG needle, keep in mind that the horizontal fibers of the cricopharyngeus are located at the level of the lower 1/3 of the cricoid cartilage, with a width of only approximately 0.75 cm [37]. The thyropharyngeus can be reached when inserting the needle at 3 cm above the point of insertion for the cricopharyngeus muscle, laterally to the flank of the thyroid cartilage [88,89]. Real-time ultrasound monitoring during EMG needle insertion can be used to view local anatomical structures and the needle [78,80,81].

Unparalleled EMG signals from the left and right sides may be detected [7,86]. For instance, the EMG pause of the cricopharyngeal muscle may be bilaterally reduced but dropped more on the paretic side compared with the contralateral side in patients with unilateral spasticity. Alternately, one side of the cricopharyngeal muscle may present greater muscle hyperactivity than the other side. Under such conditions, the injection may be performed on one side [7]. However, the side that presents abnormal EMG activity varies from one patient to another, and both sides may be affected in some cases [86].

During an endoscope-guided injection, the bulge of the cricopharyngeal muscle, or more specifically, the horizontal component of the posterior belly of the muscle [62], can be directly visualized [63] (see Figure 3 for structures viewed by flexible endoscopy). Various injection sites have been introduced, including the dorsal part [77], four quadrants [66,74], posterior part and both lateral sides [69,76], posterior and both posterior lateral walls [70], dorsomedial part and bilateral ventromedial parts [65,66], dorsomedial part and both lateral sides [75], dorsomedial part and bilateral ventrolateral parts [63,68], two dorsomedial parts [71], or the dorsomedial, ventrolateral, and ventromedial areas [64] of the cricopharyngeus/pharyngoesophageal junction. More injection sites can facilitate the spread/diffusion of the toxin to broader areas of the cricopharyngeal muscle. However, some authors performed injections only into the posterior parts of the muscle, considering that the posterior parts can be easily identified [67], and the injection into the anterior/ventral parts may affect glottic musculature and vocalization [21,24,67].

An injection under real-time ultrasound guidance can visualize the needle and local anatomical structures without receiving ionizing radiation from X-rays during CT guidance. For ultrasound scanning in a transverse or longitudinal view, the cricopharyngeus is usually viewed from the left side of the neck [29,78,79]. When using a slightly flexed neck position with a pillow under the head and head-turning 45° to the opposite side, scanning from the left neck can provide a view of the left lateral 2/3 portion, whereas scanning from the right side can provide a view of the right lateral 1/3 portion of the cervical esophagus [31]. The muscle tissues present dark/hypoechoic signals, whereas connective tissues and fat show bright/hyperechoic signals. An outermost hyperechoic layer (adventitia), a hypoechoic muscle layer, and an innermost mixed-echoic layer composed of the mucosa and submucosa of the cricopharyngeus can be viewed from outside to inside. The cervical esophagus presents five echo layers in an outside-inside order in ultrasonography, including an outermost hyperechoic layer, a hypoechoic layer of esophageal longitudinal muscle, a hyperechoic intermuscular connective tissue, a hypoechoic layer of esophageal circular muscle, and an innermost mixed-echoic layer composed of the mucosa and submucosa [9]. Considering the difference in the instruments and descriptions, four layers of the cervical esophagus are also reported [36]. Similar to a comet-tail artifact [30], during a dry swallow, the downward movement of air generates a strong echogenic appearance passing the UES lumen, which helps in identifying the location of the cricopharyngeus among nearby structures.

Guided by ultrasonography, the needle can be inserted via either a long-axis in-plane (i.e., the needle moves along the long axis of the probe/image plane) [81,83] or a short-axis out-of-plane approach [78,79] to reach the cricopharyngeal muscle. The probe may be placed on the left [78,79,81,83] or either side [81] of the neck. If the operator uses a needle approach, bypassing the thyroid gland and big blook vessels, the insertion site should be on the lateral side of the neck instead of the anteromedial side [81,83] (Figure 4). The anterior and posterior walls of the cricopharyngeal muscle [78] or longitudinally-distributed sites of the UES [83] can be injected. The dark/hypoechoic muscle layers of the UES are the target of the needle tip.

A balloon catheter guidance can be applied simultaneously with real-time ultrasonography. This procedure is performed using a dual-lumen Foley catheter with a balloon. After the catheter is inserted into the esophagus through the nose or mouth, the balloon is inflated with normal saline at a suitable volume (depending on the extent of UES narrowing). Then, the inflated catheter is gently pulled up until it is blocked by the narrowed segment [79,80,81,82,83].

The balloon is blocked at the caudal end of UES dysfunction [42,83], which can be visualized via ultrasonography (Figure 5). This is the unique merit of the procedure because the entire narrowed segment of the UES can be determined in such a manner. A barium swallow test or manometry can hardly achieve this goal. In a barium swallow study, incomplete or failure of UES opening occurred in the cranial/upper border of the narrowed UES. However, a clear caudal/lower border of UES dysfunction may not be seen via a barium test under some conditions, such as in UES opening deficits without a cricopharyngeal bar [37] or complete UES closure during swallowing. The high-pressure zone presented in a manometry test can be regarded as the length of the UES, but the relationship between the anatomical structures and the high-pressure zone may be inconsistent [37], because the spatial relation between the cricopharyngeal muscle and manometric sensors during swallowing is changing caused by laryngeal elevation [68]. Additionally, for a balloon catheter application combined with ultrasonography, the blocked inflated balloon can be visualized in real-time to guide the needle insertion towards the wall of the narrowed UES [80,81], and this process cannot be achieved by manometry, CT, or a barium study.

The volume used to inflate the balloon may be used as an index to measure the severity of UES narrowing. A volume cut-off value is not available by now, although a lower boundary of 2.5–4 mL (possibly in patients with severe UES dysfunction) [79,80,83] before receiving botulinum toxin injection has been reported. If the inflated balloon is pulled up, the segment above the blocked position (e.g., at the level of C6 [83] or C6–C7 [79]) is squeezed to various extents. The length of such a squeezed segment is shorter than that of the non-squeezed segment.

Simultaneous EMG, ultrasound, and balloon catheter guidance have been used for botulinum toxin injection [80,81]. Simultaneous EMG recording of tonic muscle activities can indicate the entrance of the needle tip into the muscle layers of the UES wall.

The dosages of botulinum toxin injection vary across different studies [62] (Table 1) and are affected by many factors, including the degree of UES narrowing (e.g., UES residual pressure measured by manometry [80]), length of UES narrowing, number of injection sites, and whether the toxin can be accurately injected into the muscle. After injection, symptoms of dysphagia might be completely or partially relieved. A second/third injection might be considered when UES dysfunction was relieved by the former injection but the symptoms reappeared after several weeks/months [24,86,89]. In the United States, botulinum toxin injection for the treatment of UES dysfunction is an off-label usage. Thus, careful selection of indications and discussion with the patient are necessary before injection.

Side effects are often mild, including belching, worsening of reflux, heartburn, increased hypopharyngea1 retention, temporary worsening of dysphagia, or dysphonia, but serious complications including pharyngocutaneous fistulas, mediastinitis, and perforation of the esophagus have also been reported [62,63,65,70].

## 6. Summary and Discussion

The UES is a special target for botulinum toxin injection because of several reasons. First, the UES musculature is thin, with a width of approximately 2.7–2.8 mm. The inner circular muscle layer, as the injection target, has approximately half of such a width. In addition, motor endplates in the cricopharyngeus and cervical esophagus segments of the UES are scattered in patterns. Moreover, the upper and lower borders of the abnormal UES segment may vary in patients and need to be determined before injection.

Precisely piercing the needle tip into the affected muscle layer, injecting more sites in the transverse plane and along the longitudinal axis of the UES (three-dimensional consideration), using botulinum toxin products with a higher spread/diffusion capacity, diluting the toxin with a larger volume, and injecting a higher dose can better block muscle contraction. Otherwise, UES dysfunction may be incompletely relieved. Moreover, the excessive effects of chemodenervation that affect adjacent muscles and hinder the anti-reflux function of the UES should be avoided.

When performing intraluminal injections under endoscope guidance, injection sites in the transverse plane can be conveniently reached in the postcricoid region. The ultrasound-guided transcutaneous injection can pierce into the anterolateral and posterior walls of the UES and reach sites along the UES longitudinal axis. Whether the needle tip pierces into the UES muscle layers can be determined via real-time ultrasound and/or EMG monitoring. Based on morphological features, the inner circular muscle layer instead of the outer longitudinal muscle layer of the UES contributes to the opening deficits, though relevant EMG studies are lacking. However, the toxin may affect the entire musculature at the injected level because either the UES wall or its muscle layer has a small width.

The upper border of the UES dysfunction can be seen in a barium swallow. Based on personal experience, after drinking as little as 1 mL of barium liquid just before injection, the lower border of barium retention in the postcricoid region is the upper border of UES dysfunction, which can be visualized via real-time ultrasound monitoring during the injection. Dual guidance with ultrasound and a blocked inflated balloon can detect the lower border of the UES narrowing.

## Figures and Tables

**Figure 1 toxins-14-00321-f001:**
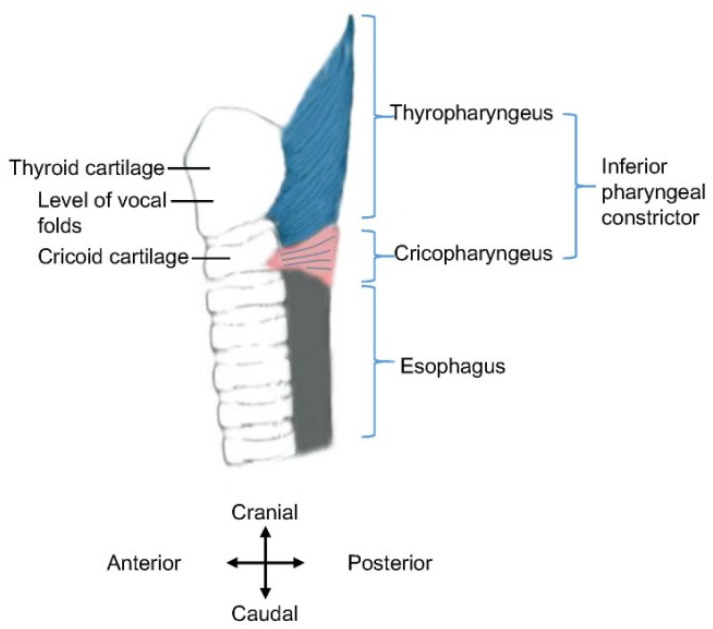
Schematic diagram of inferior pharyngeal constrictor. Anatomical structures are displayed in the sagittal plane (i.e., lateral view). The approximate location of the vocal folds is indicated.

**Figure 2 toxins-14-00321-f002:**
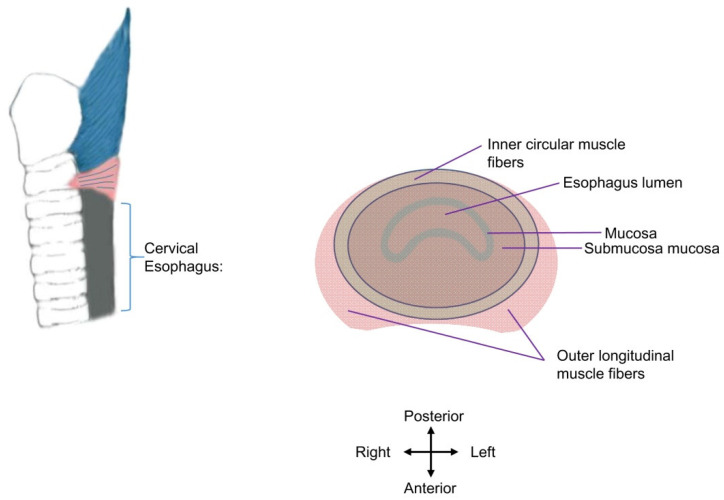
Schematic diagram of cervical esophagus muscle layers. The esophagus lumen is enclosed by the mucosa of the cervical esophagus. The inner circular layer (in grass green) and outer longitudinal muscle fibers (in pink) are presented. The adventitia of the cervical esophagus is not shown. The inner circular layer is the injection target.

**Figure 3 toxins-14-00321-f003:**
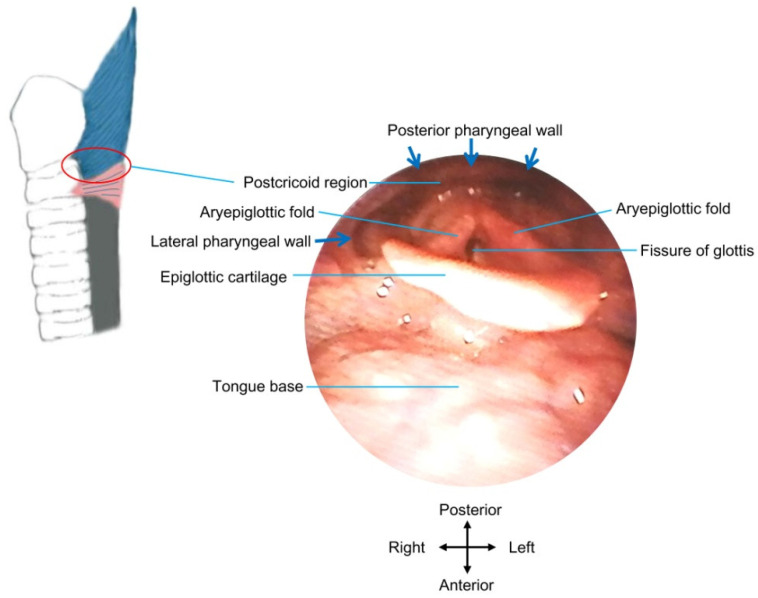
Structures under direct view by flexible endoscopy. The red circle indicates the approximate location of the postcricoid region, where botulinum toxin injection can be administrated into the UES wall under direct view by flexible endoscopy.

**Figure 4 toxins-14-00321-f004:**
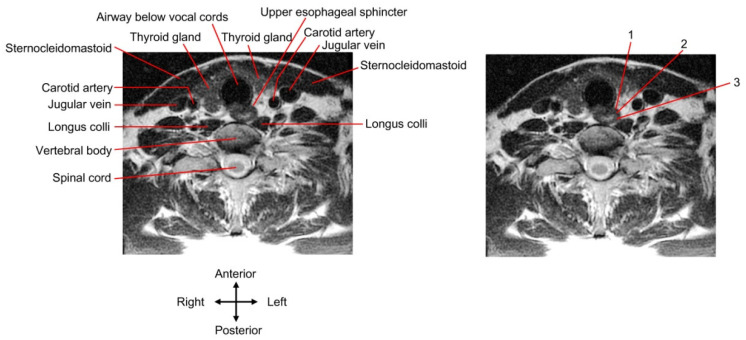
Transcutaneous injection approach. A cervical spine magnetic resonance image (MRI) of a young, healthy individual is presented. The UES slightly deviates from the midline to the left. The MRI of the (**right**) panel is the same as the (**left**) one and demonstrates three injection approaches as follows: 1, the needle route passes through the thyroid gland; 2 and 3, the needle route passes through the lateral side of the neck to bypass the thyroid gland and big blood vessels.

**Figure 5 toxins-14-00321-f005:**
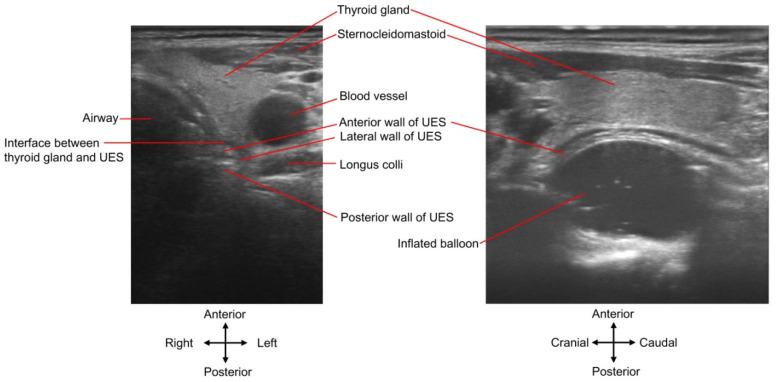
UES and inflated balloon viewed by ultrasound. The inflated balloon dilation procedure is monitored by ultrasound. The (**left**) panel is a cross-sectional demonstration. Note that the hypoechoic interface (dark zone) between the thyroid gland and the UES is not the UES wall. The (**right**) panel presents the balloon moving along the long axis of the UES. See Appendix A for dynamic display.

**Table 1 toxins-14-00321-t001:** Dose, injection sites, and guiding techniques for adults.

Literature	Toxin	Dose	Dilution Volume	Injection Sites	Guiding Techniques
[7]	Xeomin	15 or 20 U	5 U/0.1 mL	The cricopharyngeus (on the side showing greater muscle hyperactivity)	EMG
[42]	Hengli	75 U	100 U/2 mL	The cricopharyngeus muscle at three different locations	CT + balloon
[62]	BOTOX	15–100 U	0.4–0.5 mL	Horizontal component of the posterior belly of the cricopharyngeus	EMG, or laryngoscopy
[63]	Dysport	80–120 U	2.5 mL	Dorsomedial part, bilaterally ventrolateral parts	Rigid esophagoscopy + EMG
[65]	BOTOX	100 U	2.5 mL	Posterior and both lateral sides of the cricopharyngeus	Rigid laryngoscope
[66]	BOTOX	25–50 U	25 U/mL	Rigid endoscopy: 4 quadrants; flexible endoscopy: dorsomedial part and bilateral ventromedial parts	Rigid endoscopy + EMG, or flexible endoscopy
[67]	BOTOX	14–50 U	100 U/mL	Two posterolateral parts	Hypopharygoscopy
[68]	BOTOX	100 U	0.5 mL	Dorsomedially, and ventrolaterally on both sides	Nasopharyngolaryngoscopy
[69]	BOTOX	100 U	1 mL	Posterior part (40 U) and both lateral sides (30 U each) of the cricopharyngeus muscle	Flexible endoscope
[76]	BOTOX	100 U	2 mL	Posterior part (50 U) and both lateral sides (25 U each) of the cricopharyngeal muscle	Flexible endoscope
[77]	Dysport	180 U	200 U/mL	Dorsal part of cricopharyngeal muscle	Endoscopy
[79]	BOTOX	50 U	1 mL	Left side of the cricopharyngeus	Ultrasound
[80]	BOTOX	30–100 U	100 U/mL	Left or right side of the cricopharyngeus	Ultrasound + balloon + EMG
[81]	BOTOX	30 U	0.4 mL	The cricopharyngeus muscle	Ultrasound + balloon + EMG
[83]	Hengli	60 U	20 U/0.1 mL	Left side of the cricopharyngeus muscle	Ultrasound + balloon
[84]	BOTOX	100 U	2 mL	The cricopharyngeus, 2 cm above (i.e., inferior constrictor), and 1–2 cm below t (i.e., cervical esophagus)	EMG
[86]	BOTOX or Dysport	Low: 10–15 U BOTOX or 30–60 U Dysport; Intermediate: 20 U BOTOX or 80 U Dysport; High: 25–30 U BOTOX or up to 100 U Dysport	Dysport 500 U/2.5 mL; BOTOX 100 U/2 mL	One or two sides in the cricopharyngeal muscle	EMG
[88]	Dysport	60 U	30 U/2 mL	Bilateral cricopharyngeal muscle	EMG
[89]	BOTOX	20 U	10 U/2 mL	Each side of the cricopharyngeal muscle	EMG

If multiple injections were performed, the dosage, and volume of the first injection are presented.

## Data Availability

Not applicable.

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
