# Peer review of "Botulinum Toxin Injection for the Treatment of Upper Esophageal Sphincter Dysfunction"

_toxins, 2022, doi:10.3390/toxins14050321_

Round 1

Reviewer 1 Report

This is a narrative review focusing on the treatment of upper oesophageal sphincter dysfunction.

This review extensively describes the anatomy and physiology of the UES as well background knowledge and guiding techniques for botulinum usage. I commend the authors for their comprehensive study of the literature and for writing this review. 

Before publication there are several concerns:
The text is quite lengthy. Maybe some parts of the text can be omitted or summarized in tables.
I also recommend adding some figures of the described application techniques.
As e.g. Botox is not approved for esophageal usage by the FDA, it is crucial to address this fact and discuss it. What about dosage? How many I.U.s are recommended?
What about clinical effects? How often do you need to treat patients? What about side effects?
Please avoid the term "normal people". What about healthy individuals?

Author Response

Response: We really appreciate the comments and suggestions.

Main texts of the revised manuscript have been shortened and repeated contents have been deleted.

Five figures have been supplemented to present anatomical structures and described techniques.

We have replaced the word “normal people” with “healthy individuals/subjects”.

The contents on dosage, clinical effects, side effects of injection, and interval between injections have been supplemented as the last two paragraphs of Section 4. As we cited, dosages used in some studies was summarized in [62], which are quite variable and thus no recommended dose is given here. The fact that Botox is not approved for esophageal usage by the FDA has also been introduced in this part.

Reviewer 2 Report

Dear authors,

Dear, authors,

Thank you for your precious research that deserves to be published in the Toxins. 

The paper introduced specificity of the UES for botulinum toxin injection and key points for practitioner.

Even if the text is well organized and written, figures should support the meaningful information to the readers.

I strongly recommend figures with injection points targeting the muscle to be added in the revised paper. 

As well, too much redundant sentences throughout the manuscript. 

Consider shortening the text.

If this is met, I would like to give full consideration.

Author Response

Response: We really appreciate the comments and suggestions.

Five figures have been supplemented to present anatomical structures and described techniques.

These figures present the longitudinal and axial views of the UES muscle layers, nearby structures, and injection routes. Target points can be referred to Figures 2 and 3.

Main texts of the revised manuscript have been shortened and repeated contents have been deleted.

Round 2

Reviewer 1 Report

Thank you for addressing the issues with the manuscript of the review Botulinum toxin injection for the treatment of upper esophageal sphincter dysfunction.

I think the figures have improved the text.
However, I still think, that the review is too lengthy and the focus on anatomy and function is described too detailed. Also, the authors did not respect the suggestion, that tables may be used to shorten the text and improve its readability.

The sentence addressing the regulatory authorities is poor and does not contribute any content. Please state directly, that this is an off-label usage, which is completely fine, if indicated and discussed with the patient.
What about dosage? You are describing the thickness of the esophageal wall of cadaver studies in such detail, but you are not including the dosage of the various products in the toxins journal? 

In conclusion, my recommendation is that the manuscript has to be adapted to the focus of this journal.

Author Response

Response: Many thanks for the comments and suggestions.

Some introduction on anatomy and function has been deleted in Section 2 (UES anatomy and function). To improve readability, a table has been supplemented to introduce dose, injection sites, dilution volumes and guiding techniques if all these information were available in a cited study.

The following information is stated directly in the revised version: In United States, botulinum toxin injection for the treatment of UES dysfunction is an off-label usage, and careful selection of indications and discussion with the patient are necessary.

The dosage of injection can be found in Table 1 in the revised manuscript. In this table, the dosage of the various products including Botox, Dysport, Xeomin, and Hengli are presented. Several types of information are presented together for reference, considering that the selection of injection dose may be affected by these factors such as toxin productions, injection sites, dilution volumes and guiding techniques.

Reviewer 2 Report

Dear authors,

Thank you so much for the taking time revising the paper. 

I would like to have this in a accepted in present form. 

Author Response

Thank you very much for the comments and suggestions.